# Overview of the Structure–Dynamics–Function Relationships in Borohydrides for Use as Solid-State Electrolytes in Battery Applications

**DOI:** 10.3390/molecules26113239

**Published:** 2021-05-28

**Authors:** Tabbetha A. Dobbins

**Affiliations:** Department of Physics and Astronomy, Rowan University, Glassboro, NJ 08028, USA; dobbins@rowan.edu; Tel.: +1-856-256-4366

**Keywords:** solid-state battery, lithium ion battery, borohydride, closo-borane

## Abstract

The goal of this article is to highlight crucial breakthroughs in solid-state ionic conduction in borohydrides for battery applications. Borohydrides, M^z+^B_x_H_y_, form in various molecular structures, for example, nido-M^+^BH_4_; closo-M^2+^B_10_H_10_; closo-M^2+^B_12_H_12_; and planar-M^6+^B_6_H_6_ with M = cations such as Li^+^, K^+^, Na^+^, Ca^2+^, and Mg^2+^, which can participate in ionic conduction. This overview article will fully explore the phase space of boron–hydrogen chemistry in order to discuss parameters that optimize these materials as solid electrolytes for battery applications. Key properties for effective solid-state electrolytes, including ionic conduction, electrochemical window, high energy density, and resistance to dendrite formation, are also discussed. Because of their open structures (for closo-boranes) leading to rapid ionic conduction, and their ability to undergo phase transition between low conductivity and high conductivity phases, borohydrides deserve a focused discussion and further experimental efforts. One challenge that remains is the low electrochemical stability of borohydrides. This overview article highlights current knowledge and additionally recommends a path towards further computational and experimental research efforts.

## 1. Introduction

Recent advances in battery technologies have permitted the development of rechargeable lithium ion batteries with high energy density, high power density, durability, long term cycle life, and low loss in capacity with cycling. The intent of this article is to provide inspiration to undertake research to advance “beyond lithium battery concepts”, specifically towards borohydride solid-state electrolytes. However, today lithium ion batteries are dominating the market. Lithium ion batteries boast specific power densities of ~500 W/kg and specific energy densities of up to 200 kWh/kg (values reported using 50% efficiency estimates) [1]. These lithium ion batteries use an organic liquid electrolyte to ensure high ionic conduction. Safety remains a concern for these materials because of the organic materials in the electrolyte layer [2,3,4,5]. Figure 1a schematically shows currently used batteries with an organic liquid electrolyte layer to transport the Li^+^ ions. Figure 1b schematically shows the essential components of an all solid-state battery—containing a solid electrolyte layer between the electrodes. For the all solid-state lithium ion battery, it is important for Li^+^ to diffuse rapidly across the electrolyte layer. Solid-state electrolytes, which have high ionic conduction, will reduce the need for organic liquid electrolytes and will, in turn, have a large impact on further growth of lithium ion battery technologies [6]. Materials development in the electrolyte layer will revolutionize applications available for lithium ion battery use.

### 1.1. Why Study Borohydrides for Solid-State Electrolytes? The Versatility of Boron–Hydrogen Chemistry

This overview article work seeks to provide discussion germane to materials based in the borohydride family for solid-state electrolyte applications. The development of borohydrides as a class of materials for use as solid electrolytes could lead to innovations in low temperature fuel cells and safer ion conducting batteries. This class of materials has not been fully explored experimentally or computationally for use as solid-state electrolytes. A review by Matsuo and Orimo published in 2011 describes fast ionic conduction in LiBH_4_, LiNH_2_, and LiAlH_4_ systems (and their mixtures), providing a prospectus on improving room temperature ionic conduction [7]. Nevertheless, these classes of materials do not explore the full spectrum of borohydride polymorphs. The promise of borohydrides for solid-state electrolytes was also assessed in a review article by Cuan et al. in early 2019 [8]. That comprehensive review is useful, and should be considered, for a very in-depth discussion of the ionic conductivity of borohydrides and carboranes as candidate solid-state electrolyte materials.

The attractiveness of borohydrides is that they form in various molecular structures. Boron–hydrogen chemistry is versatile (forming bridge and cyclic structures), and this could lead to improved tunability in cationic conduction. For example, nido- M^+^BH_4_, closo-M^2+^B_10_H_10_, closo-M^+^B_12_H_12_, and planar-M^6+^B_6_H_6_ (with M = cations) can form ionic compounds with cations such as Li^+^, K^+^, Na^+^, Ca^2+^, and Mg^2+^. Anion structural renderings are shown in Figure 2. The closo-borane (-B_10_H_10_^2−^ and -B_12_H_12_^−^) structures have open channels for ionic conduction [9].

### 1.2. Key Technical Challenges for Borohydrides as Solid-State Electrolytes

There are several technical challenges remaining for lithium ion batteries (in general) and, in particular, for solid-state batteries. Among those are increasing ionic conduction, broadening the electrochemical stability window, alleviating dendrite formation, and achieving high energy density. Figure 3 is reproduced from the work by Lu et al. [10] and shows an assessment (in spider diagram) of how well the materials perform as battery electrolytes with respect to ionic conduction, dendrite formation, and width of the electrochemical window (which is defined by anodic reaction and electrochemical oxidation potentials) [10]. Like Lu et al. [10], many others have undertaken comprehensive engineering studies on borohydride materials, showing their promise for solid-state battery applications [6,11,12,13,14,15,16,17,18,19]. This overview article examines some of the scientific underpinnings for these performance metrics in borohydride solid electrolytes. The next sections of this overview article focus their discussions solely on ionic conduction and electrochemical stability in order to render detailed treatment of those topics (with some suggestions for experimental undertakings that would enhance the field). This article serves to inspire further research into the topic of borohydrides for solid-state battery applications.

## 2. Borohydrides as Solid-State Electrolytes

### 2.1. Foundational Work with Borohydrides in the Electrolyte Layer of Batteries

Early research utilizing borohydrides in the electrolyte layer was done with the addition of organic liquid carriers [11,20,21]. Work published by researchers at the Toyota Research Institute of North America (Ann Arbor, MI) [20,21] used organic liquid phases tetrahydrofuran (THF) and dimethoxyethane (DME) with Mg(BH_4_)_2_ and LiBH_4_ mixtures as electrolytes in operational batteries and showed reasonably high full cell electric potential and cycle life. Work using Mg(BH_4_)_2_ and LiBH_4_ was followed by use of carboranes, e.g., (C_2_BH_10_H_11_)_2_Mg, in diglyme and tetraglyme solvents as electrolytes [11,22] which demonstrated a wide electrochemical window of 3.6 V [11]. Similar data were collected in our laboratory at Rowan University in order to determine whether particle morphology is influenced by cycling. Figure 4a shows results of three cycles for voltammetry done by on solutions similar to those used in literature [20,21] on 2 M LiBH_4_ with 0.5 M NaBH_4_ in THF solution. Microstructure changes are indeed apparent before (Figure 4a) and after (Figure 4b) cycling, and these changes could play a role in limiting the cycle life of batteries formed using these materials.

Contemporaneous with work done using borohydrides inside of solvents, borohydrides in the closo-borane phases were being considered as superionic conductors for rapid ion motion through their open channel pore structures [12,23,24,25]. Researchers doing theoretical work began to suggest that the superionic conduction would make closo-boranes promising for solid electrolytes [10,12,26]. The reasons for this superionic conduction are described in the theory paper by Kweon et al. as a flat energy landscape (with many available sites for hopping), symmetry competition that ejects the cation from particular sites, and thermal reorientation of the anions [12]. Since Lipscomb first published the probable structure of B_10_H_10_^−^ in 1959 [27], there have been many studies undertaken to understand the aromatic character over 3D space in carboranes and boranes primarily by NMR and density functional theory [28,29]. Only a few crystallographic measurements have been performed on closo-boranes using diffraction [30,31,32,33,34]. Many years after the first proposed structure of the closo-borane B_10_H_10_^−^, a 2004 diffraction study by Lipscomb confirmed the Cu_2_B_10_H_10_ structure [31]. Experimental studies showed that Li_2_B_12_H_12_ has an ionic conduction of 0.1 S/cm at ~110 °C, even higher than was predicted [13]. Further, Kim et al. [14] demonstrated that with deficiencies on the Li and H sites in Li_2_B_12_H_12_, ionic conductivities can be improved by three orders of magnitude [14]. Other studies on solid electrolytes with the BH_4_^−^ allomorph have demonstrated their feasibility for competitive ionic conduction (but with a narrower electrochemical window relative to sulfide and phosphate class of solid electrolyte materials) [15,16,17,18,19,35].

### 2.2. A Survey of Ionic Conductivity in Borohydrides

Ionic conduction is considered the most important variable for solid electrolytes, and hence there have been many material developments for improved ionic conductivity [9,20,36,37,38]. Table 1 describes some experimentally and computationally determined ionic conductivities. Some traditional ionic conductors that are considered “good” in performance with respect to this measure are included in Table 1 for comparison. Addition of adducts and other modifications to borohydrides have been demonstrated to be effective at improving ionic conduction [6,39,40,41]. As an example, pure lithium borohydride (LiBH_4_^−^) has a room temperature ionic conductivity of ~10^−7^ S/cm [40]. Doping and adducts have been demonstrated to enhance ionic conductivities. These increases are realized for a variety of reasons, among which are stabilizing high ionic conductivity phases, expanding the lattice, and forming high conductivity sub-lattice pathways [40,41]. Work has been done in adding ligands, such as -NH_2_, to affect ionic conduction in borohydrides. For example, MgBH_4_ having an ionic conductivity of 10^−8^ S/cm realizes an increase in ionic conduction to 10^−5^ S/cm with the addition of an -NH_2_ adduct [39].

The conductivity for Mg(BH_4_)_2_ with an adduct of -NH_2_ reaches ionic conductivity levels of 10^−6^ S/cm at 150 °C [39,42]. This compares well with the highly researched Mg_3_(PO_4_)_2,_ which has an ionic conductivity of 10^−6^ S/cm at temperatures in excess of 400 °C. Another factor that is well known to affect ionic conduction in LiBH_4_ is crystallographic phase. When measured above the orthorhombic to hexagonal phase transition (occurring at 130 °C), a higher ionic conduction is realized in the elevated temperature hexagonal phase [15]. The lithium ion conductivity is realized in the hexagonal phase of LiBH_4_ is 10^−3^ S/cm [15]. It will be useful to develop approaches to stabilize the elevated temperature phase down to room temperature. Work done by doping LiCa_3_(BH_4_)(BO_3_), formed by decomposing LiBH_4_, Ca(BH_4_)_2_ in the presence of carbon, has demonstrated a Li^+^ conduction of 10^−5^ S/cm at room temperature [41]. The -BO_3_ molecular unit is reported to define the transport pathway for the lithium ion [41]. Additions of, Sr^2+^, Eu^2+,^ and Na^+^ have all demonstrated improved Li^+^ conductivity when added to LiCa_3_BH_4_BO_3_. The larger-sized divalent substitutions, Sr^2+^ and Eu^2+^, expand the lattice to enhance transport. The monovalent Na^+^ substituting onto Ca^2+^ O-sites also causes a lattice expansion when there is excess Li^+^ [41].

As noted earlier in this paper, the open structure of the closo-boranes (B_n_H_n_^−^) results in increased ionic conduction over the nido- BH_4_^−^ units. In fact, closo-boranes realize ionic conductivities, which are five orders of magnitude higher than nido- structures. Other work has shown that rapid cationic conduction (3 × 10^−2^ S/cm) occurs in the LiB_10_H_10_ and NaB_10_H_10_ phases [10,12]. Similarly, rapid ionic conduction is observed for the -B_12_H_12_ closo-borane phases [10,11,12,20,24]. Because of their high ionic conductivities, ease of synthesis and safety of use, for the closo-boranes should also be considered. The closo-boranes (B_n_H_n_^−^) could readily be synthesized following the resin-based ion exchange methods as described in the thesis by Blake [43], beginning with commercial triethanolamine (TEAH)-B_n_H_n_ purchased from Boron Specialties, LLC. Some borohydrides, such as borane, diborane, and pentaborane, are known to be toxic and pose environmental safety hazards and so safe handling must be considered. These are primarily in stoichiometries x < 5 in B_x_H_y_ (e.g., BH_3_, B_2_H_6_, or B_5_H_9_) and are liquid or gaseous at room temperature, toxic, and highly reactive with oxygen in air.

### 2.3. Electrochemical Stability Window of Borohydrides

The electrochemical stability window is the electric potential range over which the material is electrochemically stable. One drawback of borohydrides is their narrow electrochemical windows [10,44]. Figure 5 shows the calculated electrochemical stability ranges for common electrolytes and electrodes (relative to lithium metal at 0 V). The Cuan et al. [8] R article provides example of electrochemical stability window for a limited selection of borohydrides. A more comprehensive list relative to these materials is provided by Ceder et al. [45] and Lu et al. [10]. Most electrochemical stability window data reported in the literature has been computationally predicted. Although focused on high ionic conduction oxides, nitrides, and sulfides, Ceders et al. [44] include LiH and LiBH_4_ in electrochemical stability window calculations (Figure 5). Lu et al. [10] provide electrochemical stability window calculations for a variety of borohydrides shown in Figure 5. With the exception of CaB_12_H_12_, the electrochemical stability ranges are all below 3V. Lu et al. [10] also provide comparison to experiments in the cases of LiBH_4_ and NaBH_4_ (not shown in Figure 5). For all materials examined in that work, the measured stability window is larger than the computationally predicted electrochemical window. This is reportedly due to slow kinetics of the decomposition reaction (and product phase formation) at the cathode [10]. Furthermore, Lu et al. [10] show electrochemical windows computational predictions assuming removal (i.e., not inclusive) of B_x_H_y_ product species. Higher cathodic potentials are demonstrated in computations performed without the B_x_H_y_ species [10]. The electrochemical window of the borohydride LiBH_4_ is predicted to be low, ranging from 0.5 V to 1.9 V (vs. Li metal) [44]. Other compounds, such as Li_2_S and Li_3_PS_4,_ are stable up to 2.3 V to 2.5 V vs. Li metal, while LiH is stable down to 0 V [44]. The stability window for other borohydrides has been reported in theoretical work to be as high as 5.5 V (for CaB_12_H_12_) [10]. It will be important to establish reliable approaches to measure the electrochemical window. In practice, B_x_H_y_ product species cannot be easily removed and must be considered. In practical applications, the electrochemical windows could be far from those predicted computationally. This is an important avenue of experimental research that should be undertaken. Moreover, research which considers interlayered compounds comprised of several borohydrides as the solid electrolyte could serve the objective of increasing electrochemical window of the entire layer while isolating product phases into close proximity to one another for overcoming diffusion barriers to forming starting compounds upon reverse cycling.

### 2.4. Useful Characterization Techniques to Gain Insights on Structure-Dynamic-Property Relationships

The experimental data are determined by the most widely used approach for measuring ionic conductivity: impedance spectroscopy. Impedance spectroscopy is able to distinguish localized, non-localized, and space charge effects [46,47,48]. More useful insights are gained when additionally using characterization techniques that measure structure and dynamics. Lithium ion conductivity LiBH_4_ is 10^−3^ S/cm in the high temperature hexagonal phase [15]. To examine the stability of the hexagonal phase at temperatures lower than 130 °C, we used Raman spectroscopy to track the structural changes in the BH_4_^−^ unit with temperature on ramp up and ramp down at a slow heating and cooling rate of 5 °C/min (Figure 6). Gomes, Hagemann, et al. [49] reported that Raman spectroscopy tracks structural changes occurring to the BH_4_^−^ molecular unit upon phase transition from the orthorhombic to the hexagonal phase. Figure 6 shows Raman data collected at Rowan University. Heating LiBH_4_ to elevated temperatures shows that the B-H stretching modes (at ~2300 cm^−1^) transition from a ν_3_ splitting to a single broad peak at elevated temperature, which is consistent with a phase transition to the hexagonal phase (which occurs at 130 °C) [49]. Upon cooling, the single broad peak persists down to 113 °C, suggesting the possibility to extend the structural features associated with the elevated temperature hexagonal phase down below the phase transition [15]. Additional research could explore whether the stability of this single broad peak associated with the high temperature hexagonal phase could be further decreased towards room temperature by using additives or processing conditions to stabilize it. As well, Maekawa et al. explored lithium halides for the purpose of stabilizing the elevated temperature phase [40]. Additions of LiCl were also demonstrated to increase ionic conduction in LiBH_4_^−^ (although not by the mechanism of stabilizing the hexagonal phase) [6]. Nanostructuring LiBH_4_ has a marked effect on room temperature ionic conduction, measured at 10^−3^ S/cm [18]. The relationship between the crystallographic phase, state of the BH_4_^−^ unit (determined by techniques such as Raman spectroscopy), and ionic conductivity (using impedance spectroscopy) should be explored for both nanostructured LiBH_4_ and the LiBH_4_ having additives such as lithium halides incorporated in the materials processing. Vibrational spectroscopy and other crystallographic and structural measurement techniques should be, ideally, partnered with ionic conductivity measurements in order to gain “fine-structure” insights that could lead to novel approaches to improved properties.

Other advanced characterization techniques, such as quasi-elastic neutron scattering (QENS), which can deliver both structural information (such as H-H distances), diffusivities, and mechanism of motion (i.e., tumbling, hopping, etc.), should be considered when studying ionic mobility in borohydrides. Impedance spectroscopy was used to study superionic conduction in the Na_2_B_12_H_12_ phase. Results suggest that the high rotational mobility of the in -B_12_H_12_^2−^ anionic structures aids in the high ionic mobility of Na^+^ [24]. Furthermore, QENS studies performed by two separate teams of researchers both suggest that the motion of the B_12_H_12_^2−^ anions plays a correlated role in cationic conduction [50,51]. In another example, fundamental insights were gained using QENS, and results showed that even at temperatures as low as 4 K to 70 K, tunneling rotational dynamics for molecules as large as ammonia (NH_4_) in -B_12_H_12_^2−^ structures is possible [52].

## 3. Conclusions

In work documenting battery development by Scrosati, the timeline growth in lithium ion battery sales in various market sectors (i.e., HEV, cellular, notebooks, power tools, and camcorders) between 2000 and 2015 demonstrates the enormous potential of the technology to penetrate further into new markets [5,53]. There is a need for more research to understand key scientific barriers to the development of a new class of solid electrolyte materials, i.e., borohydrides. Ideal solid-state borohydride electrolytes will address the four scientific barriers of (i) low room temperature ionic conduction in the solid-state, (ii) formation of deleterious phases and dendrites at the electrolyte/anode interface, (iii) low electrochemical window (voltage range over which the phase is electrochemically stable), and (iv) low energy density owing to poor mechanical stability of thinner electrolytes. The development of new solid electrolytes could additionally lead to innovations in low temperature fuel cells (as well as safer ion conducting batteries). Boron–hydrogen chemistry is versatile (forming bridge and cyclic structures), and this could lead to improved tunability in cationic conduction. Materials developments require a better understanding of scientific underpinnings in order to create new materials for all solid-state battery applications. Specifically, the knowledge gap yet to be addressed surrounds our understanding of which strategies for enhancing ionic mobility (i.e., additives, dopants, mixtures, adducts, and nanostructuring) ultimately perform best at enhancing solid-state diffusion in these materials.

## Figures and Tables

**Figure 1 molecules-26-03239-f001:**
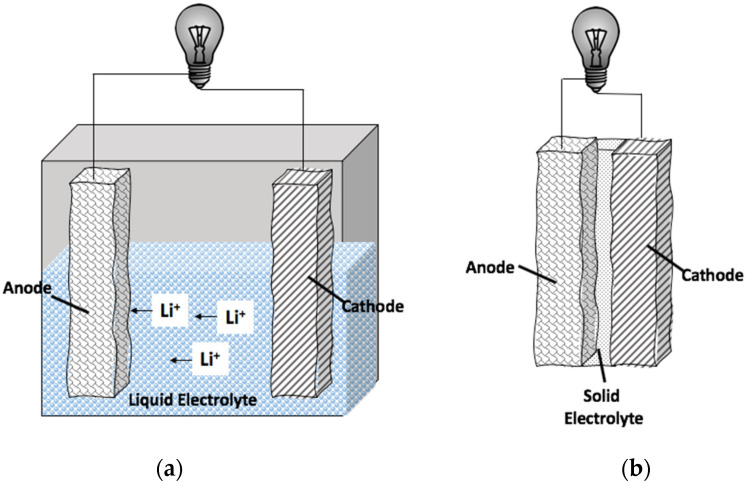
Schematics of (**a**) the current lithium ion battery containing a liquid electrolyte layer. (**b**) An all solid-state battery configuration. For the solid-state borohydride electrolyte shown in (**b**), a potential battery architecture would be comprised of LiCoO_2_ cathode and Li_4_Ti_5_O_12_ spinel anode.

**Figure 2 molecules-26-03239-f002:**
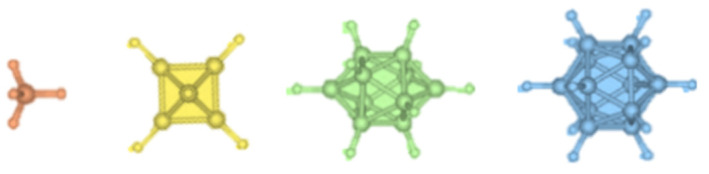
Structural renderings of (left to right) BH_4_^−^, B_6_H_6_^2^, B_10_H_10_^2−^, and B_12_H_12_^2−^ wherein the ionic radii are reported as 2 Å, 4.7 Å, 6 Å, and 5.8 Å, respectively. Image adapted with permission from [10]. Copyright permissions granted by American Chemical Society (2017).

**Figure 3 molecules-26-03239-f003:**
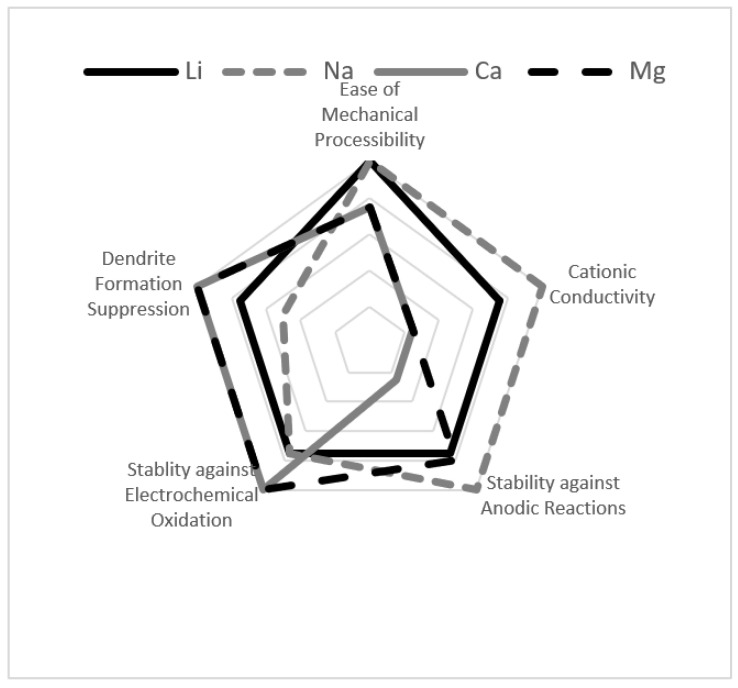
Spider diagram of key factors in solid electrolyte performance. Anodic and electrochemical oxidation stability together define the electrochemical window. Image adapted with permission from [10]. Copyright permissions granted by American Chemical Society (2017).

**Figure 4 molecules-26-03239-f004:**
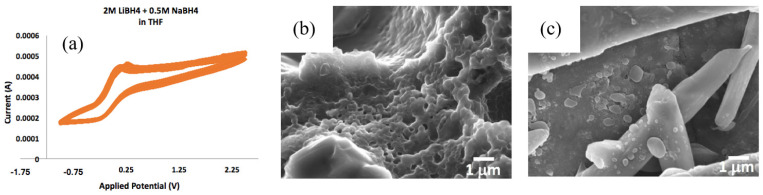
(**a**) Cyclic voltammetry done in 2 M LiBH_4_ + 0.5 M NaBH_4_ solution along with scanning electron microscopy (SEM) of borohydride powders (**b**) before and (**c**) after cycling. In solution, cyclic voltammetry drives topological changes in the hydride morphology.

**Figure 5 molecules-26-03239-f005:**
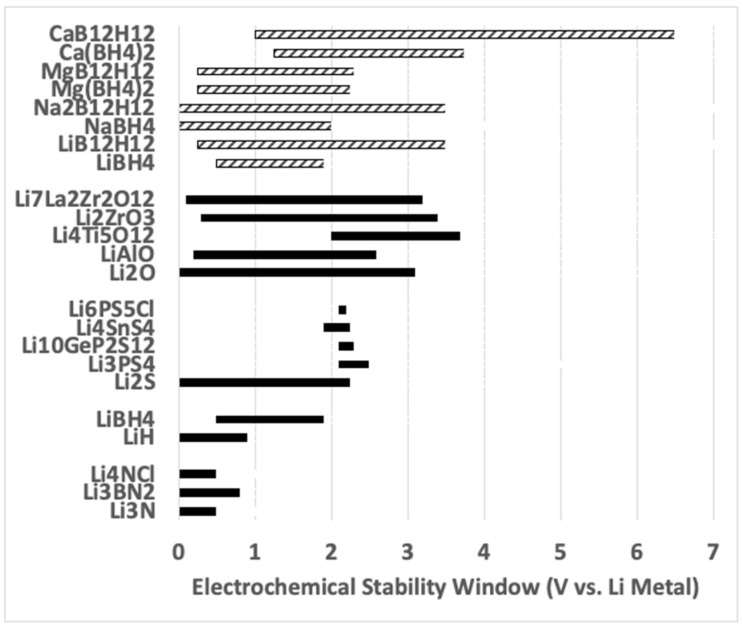
Image showing electrochemical stability windows for several potential solid electrolyte materials adapted with permission from works of Ceder et al. (in solid fill) [44] and Lu et al. [10] (in hashed fill). Copyright permissions granted by Cambridge University Press (2018) and American Chemical Society (2017), respectively.

**Figure 6 molecules-26-03239-f006:**
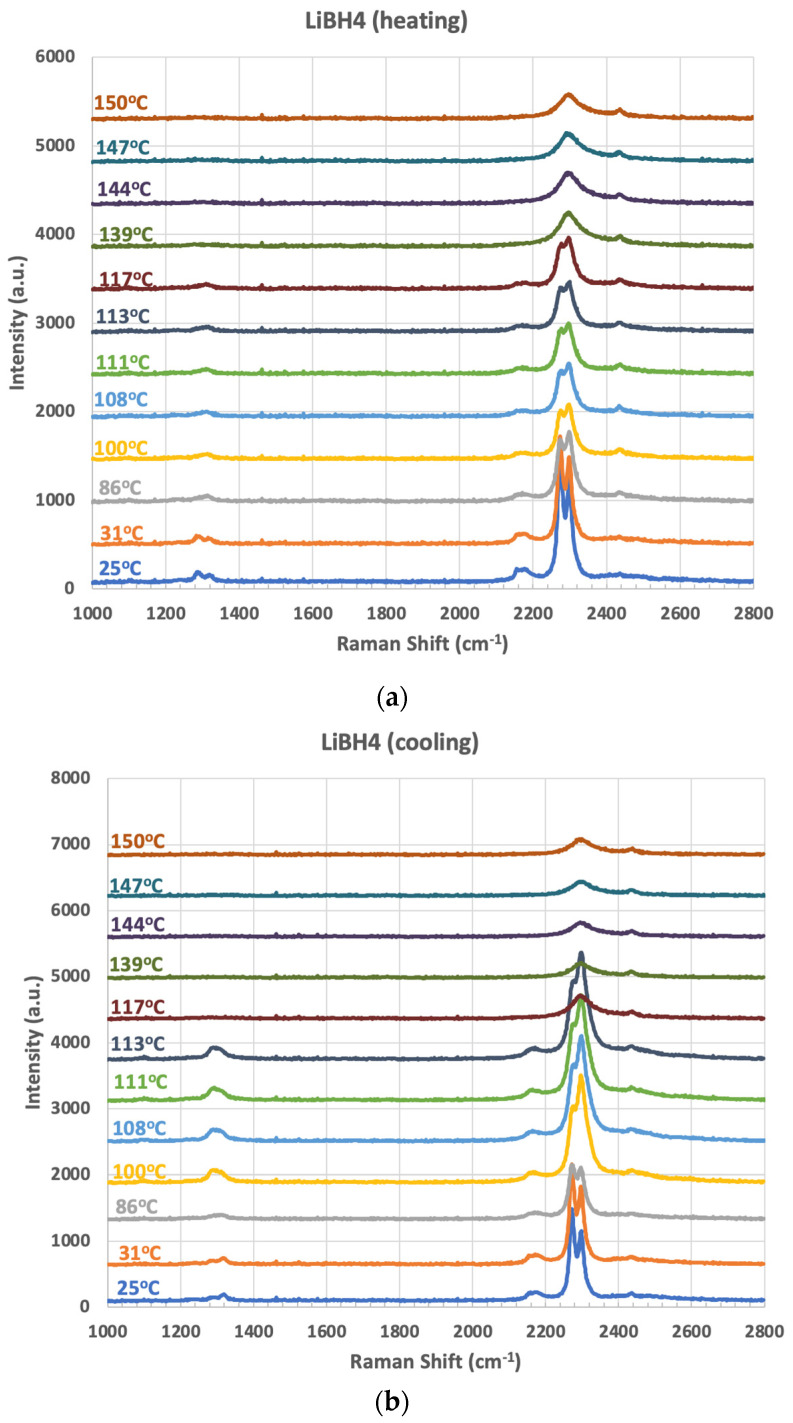
Raman spectroscopy of LiBH_4_ as a function of temperature during ramp up from RT to 150 °C (**a**) and ramp down from 150 °C to RT (**b**).

**Table 1 molecules-26-03239-t001:** Ionic mobility in solid-state electrolytes. Ionic conductivity is tuned by modification with adducts, nanoconfinement, doping, and anionic additions.

	Material	Temperature (°C)	Solid-State Cationic Conductivity (S/cm)	Reference
Traditional “Good” Ionic Conductors	Perovskites (e.g., LLZO, LLTO, LiTi(PO_4_)_3_)	RT	10^−3^ to 10^−5^	[6]
Untreated Borohydrides	LiBH_4_	107	2 × 10^−3^	[16]
	Mg(BH_4_)_2_	30	10^−12^	[39]
	LiB_10_H_10_	60	3 × 10^−2^	[10]
	NaB_10_H_10_	RT	3 × 10^−2^	[10]
Nanoconfined	LiBH_4_	RT	10^−3^	[18]
Adducts	Mg(BH_4_)_2_—NH_2_	30	5 × 10^−8^	[39]
	Mg(BH_4_)_2_—NH_2_	70	6 × 10^−5^	[39]
Cationic Doped and Anion Additions	Li_4_(BH_4_)_3_I	RT	10^−2^	[40]
	Na-doped LiBH_4_-BO_3_	RT	10^−5^	[41]
	LiCa_3_(BH_4_)(BO_3_)_2_	RT	1 × 10^−5^ to 2.5 × 10^−6^	[41]
Anionic Additions	LiBH_4_-LiX (X = Cl, Br, I)	RT	10^−4^ to 10^−7^	[6]

RT = room temperature.

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
