# Peer review of "Overview of the Structure–Dynamics–Function Relationships in Borohydrides for Use as Solid-State Electrolytes in Battery Applications"

_molecules, 2021, doi:10.3390/molecules26113239_

Round 1

Reviewer 1 Report

This reviewer sees a discrepancy between the introduction and the topic of the review article. Using the 2019 Nobel Prize and the prior work by Whittingham as an introduction then leads without warning to the sentence on line 72 “This review article work seeks to understand ionic conductivity in materials based in the borohydride family for solid-state electrolyte applications.” The introduction to the work should describe the earliest borohydride energy storage application (possibly in relation to the work by Goodenough, Whittingham and Yoshino).

This is an interesting and timely topic for a review article, but it could be improved by more thoroughly discussing the history and current state of the field. In its current version this reviewer finds the discussion on state-of-the-art is very cursory for a review article. Additionally, the choice of figures does not seem optimal. While the legend for Figure 4 is probably simply missing a word and the citation (Figure 4. (a) Cyclic voltammetry done in 2M LiBH4 + 0.5M NaBH4 solution along with scanning electron microscopy (SEM) of the borohydride powders (b) before and (c) after [CYCLING?]. In solution, cyclic voltammetry drives topological changes in the hydride morphology. [reference?].), the importance on Figure 5 is not apparent. It should be combined with ion conduction data to illustrate the structure-property correlation. The author makes it clear that section 2.3.3. is based on a hypothesis, but it would be beneficial to include either more background to clarify the origin of this hypothesis or possibly include a Figure, or both.

Author Response

1-1.  I have removed the introductory section which describes the 2019 Nobel Prize and prior work by Whittingham— and began the introduction with discussion of the lithium ion battery only. 

1-2.   The manuscript has been entirely re-shaped to address the concern about the history and current state.  The author recognized the error in writing and now the manuscript is more streamlined within the following outline:

1. Introduction

1.1.  Why Study Borohydrides for Solid-State electrolytes?  The versatility of Boron-Hydrogen Chemistry

1.2.  Key Technical Challenges for Borohydrides as Solid-State Electrolytes

2. Borohydrides as Solid State Electrolytes

2.1.  Foundational Work with Borohydrides in the Electrolyte Layer of Batteries

2.2. A Survey of Ionic Conductivity in the Borohydrides

2.3.  Electrochemical Stability Window of the Borohydrides

2.4. Useful Characterization Techniques to Gain Insights on Structure-Dynamic-Property Relationships

3. Conclusions

Within section 2, the “Foundational Work” section provides the historical context.  The other sections, 2.2 and 2.3 describe the current status.  Section 2.4 makes some suggestion of how advanced characterization tools could benefit the state of the field.

1-3.  The cyclic voltammeter data is original— and there is no reference to place here.  These are observations— and the author makes the suggestion that “these changes could play a role in limiting the cycle life of batteries formed using these materials.”

1-4.  Figure 4 was missing the word “cycling”.  It is placed there.

1-5.  To add meaning and context to figure 5 (now Figure 6), I completed the data set with the “ramp down” from 150C to room temperature.  There, the significance of a hysteresis in the phase transition is observed.  This is important because the elevated temperature phase has higher ionic conductivity.  If it could be stabilized to lower temperatures (and indeed to room temperature), this would represent an advance in the field.  Likewise, the fine structure of the B-H stretching mode (from sharp doublet peak to a broad peak) could be the signatory feature for higher ionic conduction.  My suggestion is that this should be examined more closely— and further, I suggest that there could be means to stabilize this high temperature phase both chemically and micro structurally.

1-6.  Section 2.3.3. titled “Alleviating Dendrite formation and Achieving High Energy Densities” was removed— as it was highly speculative.  As well, the additions of suggested experiments and advanced characterization techniques values of the review article on those points.  Further, Although four factors are considered key to electrolyte performance: (i) high ionic conduction, (ii) broad electrochemical window, (iii) high energy density, and (iv) resistance to dendrite formation.  It is now stated in the introduction, “This review article focuses solely on ionic conduction and electrochemical stability.”  And, of course, the sections comprising ionic conduction and electrochemical window were re-written for more clarity. 

1-7.  Thank you for carefully detailing these writing problems in the earlier manuscript.  The latest version has now been completely re-written to address these issues/concerns.  I hope that on review of the latest version, you will find added value in this review.   The utility of this manuscript is that it makes logical and literature-motivated suggestions for what would be useful experimental data throughout.

Reviewer 2 Report

“The goal of this article is to review crucial breakthroughs in solid-state ionic conduction in borohydrides for battery application”. That is firs sentence of the abstract. In reality, this article is a gathering of notes of literature without any logic. There is no line to follow and title of the sections often has nothing to do with the text.

The first section, introduction, start from 1975 and gives a overview on the Li-ion batteries development, which is not bad. Nevertheless, the necessity of the research on solid electrolyte is not clearly introduced.  

For the second section, the author starts with a comparison of different families of solid electrolytes but very confusing statement. Then the author talks about an earlier review by Matsuo and Orimo (should be much clear) but the reference is not given.

The section 2.1 talk about the versatility of Boron-hydrogen chemistry, we would expect crystal structure which is responsible for the high ionic conductivity. Section 2.2 entitled recent advances in borohydrides and their promise for solid state ionic conduction, bur the text is exclusively talk about borohydrides in liquids carries.

Section 2.3, the “body” of the article devoted to the “technical challenges” of solid state electrolytes: ionic conduction, electrochemical stability window and alleviating dendrite formation”. The section 2.3.1 for ionic conduction, there some description of the Li-ion conductivity of LiBH4 and its substituted compounds. But the description is very brief and so confusing while next paragraph the author details the Blakes thesis start with a product from company LLC. And so on and so forth…

To conclude, people learn things and get an overview on a research field by reading a review. The present paper is far to fulfill this purpose and I would reject the paper.

Author Response

2-1. The manuscript has been entirely re-shaped to address the concern about the writing.  The author recognized the mixing of various topics and now the manuscript is more streamlined within the following outline:

1. Introduction

1.1.  Why Study Borohydrides for Solid-State electrolytes?  The versatility of Boron-Hydrogen Chemistry

1.2.  Key Technical Challenges for Borohydrides as Solid-State Electrolytes

2. Borohydrides as Solid State Electrolytes

2.1.  Foundational Work with Borohydrides in the Electrolyte Layer of Batteries

2.2. A Survey of Ionic Conductivity in the Borohydrides

2.3.  Electrochemical Stability Window of the Borohydrides

2.4. Useful Characterization Techniques to Gain Insights on Structure-Dynamic-Property Relationships

3. Conclusions

2-2.  The introduction of lithium ion batteries was removed to focus more so on the borohydride materials issues.

2-3.  The reference to Matsuo and Orimo is now given.  This was a typo.

2-4.  Within the outline above, the “Foundational Work” now contains the class of borohydrides in solvents (glyme, tetraglyme, etc.).  The section titled “A Survey of Ionic Conductivity” now comprises the classes of solid state borohydrides (grouped with first a discussion of nido followed by a discussion of closo boranes).

2-5.  Thank you for carefully detailing these writing problems in the earlier manuscript.  The latest version has now been completely re-written to address these issues/concerns.  I leave n the discussion of synthesis— but it comes behind the discussion of closoborane ionic conduction.  By removing the Raman and QENS discussion to an entirely new section (section 2.4)— the “weaving together” of seamingly separate thoughts goes away.  That section now has its own motivation— to suggest advanced characterization tools to use for deeper insights into phenomena in the borohydrides.

2-6.  I hope that on review of the latest version, you will find value in this review.   The utility of this manuscript is that it makes logical and literature-motivated suggestions for what would be useful experimental data throughout.

Reviewer 3 Report

The manuscript is a rather weak overview dealing with the role of borohydrides as solid state electrolytes in ionic batteries. I regret not to consider this manuscript suitable for publication in Molecules, due to several critical issues which are listed below:

1) During reading the manuscript, the mail feeling a reader gets is that the overall architecture of the compact review is not well balanced, as some side topics are treated too in detail, while others more relevent are only mentioned or ignored. For example, the emphasis given at the beginning on the Li ion batteries hystory is clearly out of place, as the article main focus should be the research on alternatives to conventional liquid electrolytes. A similar introduction could be justified if the manuscript lenght was much more than 9 pages... On the contrary, the connection between the role of solid electrolytes and the need of obtaining high energy densities is not obvious and it is not explained in sufficient detail in the text.
A reshape of the overall structure of the manuscript is strongly required at this stage.

2) The manuscript has many typos within the text and generally appears rough, superficial and hasty written.
Some examples:
Line 7: the complete author affiliation is missing.
Line 136: In table 1, cationic conductivities reported is probably wrong (how could be [Mg(BH4)2 – NH2] conductivity lower at 150°C than at RT?) and scientific notation is used in wrong way (please always use scientific notation to express ion conductivity within the text.
Line 180: it is not clear why in Figure 6 some labels on the left side refer to more than one potential window. Author should better explain in the caption how to correcty interpret the figure.
Reference section: many references are not correctly cited, i.e. ref. 7, 27, 38, 38...
I suggest a deep revision of the manuscript before resubmission.

Other less critical issues, which should be fixed:

2) Author spent a lot of attention on Mg2+ ion conductors with poor performance (i.e. Mg2(PO4)2 and Mg(BH4)2), but research on Mg ion batteries is still at early stage. In my opinion more focus should be spent on Li and Na ion solid electrolytes, with more details also concerning the synthesis procedures.

3) I do not well understand why the author chose to show Raman spectroscopy data to present the well-known first order transition of LiBH4, while diffraction, conductivity and/or Li NMR data would be much more meaningful in a review (see for example ref. Adv. Energy Mater. 2011, 1, 161–172) . In my opinion Figure 5 is not particularly meaningful in this context.

4) In Line 160, discussion on QENS investigation on quantum tunneling dynamics of ammonia molecules is NOT an evidence of Li or Na ionic diffusion in closo-boranes and in my opinion it is out of place in this manuscript.

5) Line 210: Li4Ti5O12 is an ANODE material and not a cathode for Li-ion batteries.

Author Response

The author appreciates the comments or reviewer #3.   Below are an outline of changes made in direct response to these comments.

3-1.  The longer introduction containing the history of lithium ion batteries was removed.  The author thought it served as motivation to put a similarly concerted effort into solid state electrolytes for lithium ion batteries. 

3-2.  In order to give the manuscript more balance, each section is made nearly equal (by addition of section headings). The manuscript is more streamlined within the following outline:

1. Introduction

1.1.  Why Study Borohydrides for Solid-State electrolytes?  The versatility of Boron-Hydrogen Chemistry

1.2.  Key Technical Challenges for Borohydrides as Solid-State Electrolytes

2. Borohydrides as Solid State Electrolytes

2.1.  Foundational Work with Borohydrides in the Electrolyte Layer of Batteries

2.2. A Survey of Ionic Conductivity in the Borohydrides

2.3.  Electrochemical Stability Window of the Borohydrides

2.4. Useful Characterization Techniques to Gain Insights on Structure-Dynamic-Property Relationships

3. Conclusions

3-3.  The entire manuscript was re-shaped for better read-ability.

3-4.  The usefulness of this manuscript is in the suggested experiments it makes throughout.  This is now a bit more apparent with the addition of a section heading for the Raman and QENS suggested experimental techniques.   That section is called: “Useful Characterization Techniques to Gain Insights on Structure-Dynamic-Property Relationships”.  Removing discussion of QENS and Raman from the ionic conductivity section (and giving its own section)— substantially shortened the ionic conductivity section making it more balanced with the section on electrochemical stability window. 

3-5.  Although four factors are considered key to electrolyte performance: (i) high ionic conduction, (ii) broad electrochemical window, (iii) high energy density, and (iv) resistance to dendrite formation.  It is now stated in the introduction, “This review article focuses solely on ionic conduction and electrochemical stability.”  And, the sections comprising ionic conduction and electrochemical window were re-written for more clarity. 

3-6.  Figure 6 (is now Figure 5) was corrupted in transfer from Microsoft Excel into Microsoft Word (for this writing).  The corruption removed some of the labels on the left.  The corrected figure is now inserted.

3-7.  The reference section has been carefully examined.  My reliance on a reference manager software caused the errors.

3-8.  Thank you for your comments regarding the magnesium materials with poor performance.  Respectfully, I have left these inside of the review for completeness in the discussion of borohydride anionic materials (with a variety of cations).

3-9.  I have added the reference you suggest by Matsuo and Orimo (2011).  To add meaning to figure 5 (and context), I completed the data set with the “ramp down” from 150C to room temperature.  There, the significance of a hysteresis in the phase transition is observed.  This is important because the elevated temperature phase has higher ionic conductivity.  If it could be stabilized to lower temperatures (and indeed to room temperature), this would represent an advance in the field.  Likewise, the fine structure of the B-H stretching mode (from sharp doublet peak to a broad peak) could be the signatory feature for higher ionic conduction.  My suggestion is that this should be examined more closely— and further, I suggest that there could be means to stabilize this high temperature phase both chemically and micro structurally.

3-10.  Because the QENS is now in an entirely new section — suggesting it be used to augment simple ionic conduction by impedance spectroscopy— I respectably suggest that the dynamics of ammonia in the closoborane structure discussion is now relevant within this new context

3-11.  The typo of cathode was removed and replaced with anode (when referring to Li4Ti5O12).

3-12.  A deep revision to the manuscript was carefully made over many hours. Thank you for carefully detailing these writing problems in the earlier manuscript.  The latest version has now been completely re-written to address these issues/concerns.  I hope that on review of the latest version, you will find added value in this review.   The utility of this manuscript is that it makes logical and literature-motivated suggestions for what would be useful experimental data throughout.

This manuscript is a resubmission of an earlier submission. The following is a list of the peer review reports and author responses from that submission.

Round 1

Reviewer 1 Report

The article reports about the structure-dynamics function relationships in borohydrides for solid state electrolytes for batteries.

The paper is submitted as a review but I think that it is very far from this classification. It is too short, any important information are present in the paper.

The paper has many problems:

The English language. It is so poor that many sentences are completely unclear and the reading is very difficult. The paper lacks of a clear project. The topic should be about borohydrides, but many other subjects are present and discussed so confusing the reader. It lacks of a clear exposition of results, of a logic conduction of the chosen topic. A reader not confident with this subject after the article reading has only confusion in the mind, nothing is clarified and nothing is conveyed to the reader.

In conclusion, the paper is badly written and conceived and does not improve the state of the art of borohydrides.

Reviewer 2 Report

The author gives a brief summarization on the development of borohydride electrolyte for battery applications. According to the authors’ discussion, my impression is that borohydride is a small research field that may not deserve a review work for the time being. In addition, the author only summarizes relating reports on borohydrides electrolyte without providing insights on the structure-dynamics-function relations and critical perspectives. Furthermore, this material system is not likely to be used for practical applications in the near future. Therefore, I could not support the publication of the manuscript on Molecules.

Reviewer 3 Report

The manuscript by Dobbins aims to review the progress of borohydride compounds as solid-state electrolytes in batteries. The author selects a trendy topic in batteries, but unfortunately, the article is highly rough and seems unfinished. As a review article, it fails to provide readers with comprehensive understandings of borohydride-based solid-state battery electrolytes (elaborated below). The current manuscript is unsuitable for publication.

There is too much irrelevant information that could distract readers. For example:

1) The first paragraph of the main text articulates the 2019 Nobel Prize in Chemistry. Although this content is associated with rechargeable batteries, it has no relationship with borohydride-based solid-state electrolytes or even battery electrolytes.

2) Lines 104-105: The experimental result was obtained with a liquid electrolyte (THF solution). This characteristic deviates from the topic of solid-state electrolytes.

3) Lines 206-211: The correlation between the discussion on Li4Ti5O12 and borohydride electrolytes is elusive.

In contrast, vital information to help readers grasp a comprehensive understanding of the review topic is all missing. This information includes but not limits to necessities of solid-state electrolytes, detailed structures of different borohydride molecules, relationships between structure and electrochemical behaviors, and ion-conduction mechanisms (e.g., superionic conduction). Additionally, the article lacks a clear plot that integrates all the discussed materials into a solid piece.

The review is far from conclusive. Through a quick literature survey, I have found at least the following relevant articles that are overlooked by the author:

Angew. Chemie 2014, 53, 3173-3177

Appl. Phys. Lett. 2007, 91, 224103

Adv. Energy Mater. 2017, 7, 1700294

Power Source 2015, 276, 255-261

APL Materials 2014, 2, 056109

Appl. Phys. Lett. 2007, 91, 224103

...

In all, there are no chances that the current manuscript is publishable after a major revision. For consideration of publication, the author needs to rewrite and re-submit the article.

Reviewer 4 Report

The review article by Dobbins, titled “Overview of the Structure-Dynamics-Function Relationships in Borohydrides for use as Solid State Electrolytes in Battery Applications” promises to review crucial breakthroughs in solid-state ionic conduction in borohydrides. However, upon reading the manuscript, this reviewer finds very little attention paid to the aspect of structure-dynamics-function relationships stated in the title.

Only in the latter part of section 2.3.1. (lines 168-181) does the author discuss the cause for an observed phenomenon. Earlier on, in the same section, the reader is left with the question “why?” after reading “Pure Lithium borohydrides (LiBH4-) has a room temperature ionic conductivity of 139 ~10-7 S/cm. Closoboranes (BnHn-) have ionic conductivities which are five orders of magnitude 140 higher.” (lines 139-141).

While it is laudable, how the author put the manuscript in a timely context by describing the work that led to the 2019 Nobel Prize in Chemistry, it would have been helpful to put the described research on borohydrides more strongly in the comparative context of other solid-state electrolyte materials, rather than utilizing the comparison to liquid organic electrolyte batteries (Figure 1).

The paragraph about LTO (lines 206-211) seems misplaced. Should this have been part of the conclusion section?

Section 2.3.3 on “Alleviating Dendrite formation and Achieving High Energy Densities” seems written in a different style. Here the author “suggests” an approach to prepare solid-state electrolyte pellets. Not to downplay the importance of the described research, but there is a shift in writing perspective here which is disruptive. Furthermore, dendrite formation and a lack of high energy density are separate issues in battery performance.

The review describes a rapidly moving field, but unfortunately crucial literature from the last year has not been cited. For example, the review article published by Cuan et al. in October 2018 (https://doi.org/10.1002/adma.201803533) on the same topic, should be acknowledged and this review should make an attempt to differentiate itself from it; the work by Maniadaki and Lodziana (Phys. Chem. Chem. Phys., 2018,20, 30140-30149) could provide guidance for future directions of the field; to name just two examples. The author should consider including further research findings in her reviewed material.

As a side note, acronyms need to be defined at first use, e.g. QENS in line 156.